# Gastric Cancer in the Era of Epigenetics

**DOI:** 10.3390/ijms25063381

**Published:** 2024-03-16

**Authors:** Grigorios Christodoulidis, Konstantinos-Eleftherios Koumarelas, Marina-Nektaria Kouliou, Eleni Thodou, Maria Samara

**Affiliations:** 1Department of General Surgery, University Hospital of Larissa, University of Thessaly, Biopolis Campus, 41110 Larissa, Greece; gregsurg@yahoo.gr (G.C.); kostaskoumarelas@gmail.com (K.-E.K.); marinakouliou@gmail.com (M.-N.K.); 2Department of Pathology, Faculty of Medicine, School of Health Sciences, University of Thessaly, Biopolis Campus, 41110 Larissa, Greece; ethodou@uth.gr

**Keywords:** gastric cancer, epigenetics, histone modifications, DNA methylation, non-coding RNAs, miRNAs, epidrugs

## Abstract

Gastric cancer (GC) remains a significant contributor to cancer-related mortality. Novel high-throughput techniques have enlightened the epigenetic mechanisms governing gene-expression regulation. Epigenetic characteristics contribute to molecular taxonomy and give rise to cancer-specific epigenetic patterns. *Helicobacter pylori* (Hp) infection has an impact on aberrant DNA methylation either through its pathogenic CagA protein or by inducing chronic inflammation. The hypomethylation of specific repetitive elements generates an epigenetic field effect early in tumorigenesis. Epstein–Barr virus (EBV) infection triggers DNA methylation by dysregulating DNA methyltransferases (DNMT) enzyme activity, while persistent Hp-EBV co-infection leads to aggressive tumor behavior. Distinct histone modifications are also responsible for oncogene upregulation and tumor-suppressor gene silencing in gastric carcinomas. While histone methylation and acetylation processes have been extensively studied, other less prevalent alterations contribute to the development and migration of gastric cancer via a complex network of interactions. Enzymes, such as Nicotinamide N-methyltransferase (NNMT), which is involved in tumor’s metabolic reprogramming, interact with methyltransferases and modify gene expression. Non-coding RNA molecules, including long non-coding RNAs, circular RNAs, and miRNAs serve as epigenetic regulators contributing to GC development, metastasis, poor outcomes and therapy resistance. Serum RNA molecules hold the potential to serve as non-invasive biomarkers for diagnostic, prognostic or therapeutic applications. Gastric fluids represent a valuable source to identify potential biomarkers with diagnostic use in terms of liquid biopsy. Ongoing clinical trials are currently evaluating the efficacy of next-generation epigenetic drugs, displaying promising outcomes. Various approaches including multiple miRNA inhibitors or targeted nanoparticles carrying epigenetic drugs are being designed to enhance existing treatment efficacy and overcome treatment resistance.

## 1. Introduction

Gastric cancer (GC) ranks fourth in terms of cancer-related mortality and stands as the fifth most prevalent cancer worldwide. There is a pronounced gender disparity, with males experiencing a two-fold higher incidence rate compared to females [1]. While the incidence and mortality rates tend to decline in developed nations, there is evidence of rising incidence rates among younger individuals (under the age of 50) [2]. Given the intricate nature of the GC pathophysiology, individuals with elevated GC risks should be given extra care while monitoring. Age, gender, and ethnicity comprise pivotal risk factors for GC development augmented by factors such as obesity, lifestyle, gastrointestinal microbiota, as well as *Helicobacter pylori* (Hp), and Epstein–Barr virus (EBV) infections [3,4].

Gastric cancer is characterized by intestinal and diffuse-type adenocarcinomas. Lately, prompt technological advancements have allowed for an understanding of the molecular mechanisms that underlie the main histological subtypes. Genome-wide association studies and transcriptomic analyses build a combined histological–molecular approach leading to molecular taxonomy [5].

The Cancer Genome Atlas (TCGA) research network team, based on sequencing data, classified GC into four molecular subtypes: chromosomal instability (CIN), Epstein–Barr virus (EBV), microsatellite instability (MSI) and genomically stable (GS) [6]. Similarly, the Asian Cancer Research Group (ACRG) Network team implemented a transcriptome classifier to identify molecular subtypes including microsatellite instability (MSI), microsatellite stability/Epithelial-to-Mesenchymal Transition (MSS/EMT), MSS/TP53 active, and MSS/TP53 inactive [7]. So far, several studies have described molecular subtyping methodologies for GC, employing high-throughput profiling and multi-omics platforms that encompass genomic, proteomic, and epigenetic features [8,9]. Li et al. (2023), using multi-omics data and integrated optimum algorithms, proposed an advanced molecular classification system identifying three GC subtypes based on mRNA, microRNA, and DNA methylation data. Subtype 1 correlates with favorable outcomes and a high mutation rate of ARID1A and *PIK3CA* mutations, whereas subtypes 2 and 3 are linked with adverse prognostic outcomes exhibiting *TP53*, *APOA1*, and *CDH1* mutations, respectively [10].

Beyond TCGA and ACRG classifications, several other classifications have been proposed. Recently, Weng et al. (2023) suggested an epigenetic-based classification. According to their proposal, the analysis of 1521 GC cases from GEO and TCGA databases in combination with miRNA-expression and DNA-methylation profiles led to four molecular GC clusters. The C1 cluster is related to the cell cycle, DNA replication and MSI and is accompanied by a better prognosis. In contrast, the C4 cluster involves immune-related processes, microsatellite stability, and TP53 mutations and relates to high-grade carcinomas and poor prognosis. Clusters 2 and 3 seem to correlate to a moderate prognosis, and advanced stages and are related to histone and DNA alterations, signaling pathways and immune invasion [11].

Epigenetic alterations exert regulatory control over gene expression without altering DNA sequence. Numerous epigenetic mechanisms, encompassing DNA methylation, histone modifications, chromatin regeneration, and miRNA interference, allow transcriptional control through regulatory proteins. Gene expression, DNA repair, and cell growth are influenced by epigenetic modifications. Possible malfunction of these mechanisms can result in the initiation of carcinogenesis. Hp infection is prevalent in over 80% of GC cases, with the resultant inflammatory milieu generated affecting both epigenetics and signaling pathways [12].

In this narrative review, we aim to present the currently available literature and enlighten the main epigenetic mechanisms that interplay in gastric cancer development, including potential biomarkers for diagnosis, prognosis, and treatment in clinical settings.

## 2. Methods

PubMed, Cochrane Library, Embase, and Google Scholar were initially searched to retrieve studies reporting data on epigenetics from 2001 to the present day. The following Medical Subject Heading [MeSH] terms were used alone or matched by the logical operators “OR” or “AND” in all possible combinations to obtain the maximal number of articles: “Gastric Cancer”, “Epigenetics”, “Stomach adenocarcinomas”, “Histone modifications”, “DNA methylation”, “miRNAs”, “Long non-coding RNAs”, “circular RNAs” “Epigenetic changes”. We excluded repetitive as well as non-English studies and we focused mainly on studies conducted after 2018. After an initial title and abstract screening, a full-text copy of each article was retrieved. Each relevant article was subsequently reviewed, and 197 representative scientific papers were finally selected.

## 3. Epigenetics Background (DNA Methylation, Histone Modifications, and Non-Coding RNAs)

Epigenetic alterations are evident in both early and advanced stages of gastric carcinomas. The rising interest in these epigenetic modifications aims to shed light on the underlying physiology of gastric cancer and unveil novel potential targets for precision medicine. Afterwards, we will highlight the main epigenetic mechanisms and their interplay during GC tumorigenesis (Figure 1).

### 3.1. DNA Methylation

Cytosine–Guanine-sequence rich (CpG) islands located within the promoter gene region play a crucial role in gene-expression regulation. DNA methyltransferases (DNMTs) catalyze the formation of 5-methylcytosine (5mC) within CpG dinucleotides. It is well established that the methylation of CpG islands within secondary promoter domains can be accelerated by factors such as aging, viral infections, and chronic inflammation. As previously reported, the analysis of the de novo methylation in gastric cell lines, exhibiting varying degrees of CpG-methylated islands, revealed elevated de novo methylation rates in cell lines containing numerous CpG methylated sites. This observation reinforces the concept that random methylation stimulates extensive CpG methylation, thus promoting gene silencing [13,14].

Previous studies have also documented the detection of aberrant methylation in *DAPK*, *E-cadherin*, *GSTP1*, *p15*, and *p16* genes in paired-tissue and serum samples of GC patients. This underlines the potential utility of analyzing methylation status for assessing and monitoring gastric cancer [15,16].

The TCGA group, conducting a molecular analysis of gastric carcinomas, has identified two subgroups characterized by elevated methylation levels. These subgroups, referred to as gastric CpG-island methylator phenotypes (CIMP) and EBV-positive tumors, including the MSI subtype, exhibit unique methylation patterns [6].

Genetic and epigenetic alterations are the key elements of *CDH1*-suppressor gene depletion, evident in both intestinal and diffuse gastric carcinomas. *CDH1* hypermethylation arises early during GC development and is detectable in nearly 50% of hereditary, diffuse gastric carcinomas (Table 1) [17,18].

DNA-mismatch-repair (MMR) pathway genes play a crucial role in preserving genomic stability across various cancers, including sporadic GCs. As previously demonstrated, the methylation of both *MLH1* and *MLH2* promoters correlates to the onset and progression of GC. In patients with surgically resectable gastric carcinomas, *MLH1* promoter methylation is associated with a favorable prognosis, whereas its absence highly correlates with tumors displaying MSI [6,19]. However, in early-stage gastric carcinomas exhibiting a papillary phenotype, microsatellite instability has been described, driven by *MLH1*-promoter hypermethylation [34].

The *CDKN2A* gene, which mediates in cell cycle arrest, is frequently methylated in GC and other gastrointestinal malignancies. Its presence in precancerous gastric lesions, associated with Hp and EBV infections, suggests its potential involvement in GC development [20,21]. DNMTs, the enzymes responsible for directing methylation processes, are notably upregulated in gastric carcinomas. As previously documented, the DNMT1 enzyme’s expression is linked to GC risk and unfavorable prognosis, particularly in stage-III and -IV GC patients [35]. The expression of the DNMT enzyme can also be influenced by the release of oncogenic proteins and inflammatory responses mediated by tumor-associated macrophages (TAMs) [36]. The *APC* gene also regulates the expression of the DNMT1 enzyme via the APC/β-catenin/TCF pathway. This hypothesis is also strengthened by the frequently observed hypermethylation of the *APC* promoter in GC patients, suggesting a possible mechanism for its inactivation [37].

### 3.2. Histone Modifications

Histones are responsible for nucleosome formation, the primary structural unit of chromatin. Histone modifications encompass a range of post-translational effects on synthesized proteins. Modifications that occur on specific amino acids (lysine, arginine, serine, and threonine), including acetylation, methylation, phosphorylation, and ubiquitination, have been described. Recently, processes like sumoylation, butyrylation, lactylation, succinylation, and crotonylation have also been added to the already-known histone modifications [38,39,40,41]. The majority of these modifications are evident in gastrointestinal tumors, potentially offering new insights into the detection and management of gastric carcinomas.

Histone methylation typically targets lysine or arginine residues in H3 and H4 proteins. The methylation of different amino acid residues is achieved through the interaction of Histone methyltransferases (HMTs) and Histone demethylases (HDMs), resulting in either gene activation or silencing [42]. The prognostic significance of EZH2 and H3k27me3 protein overexpression in gastric cancer has been elucidated, demonstrating a negative correlation between H3K27me3 levels and overall survival in GC patients [43]. Cytotoxin-associated gene A (CagA), a virulent component of Hp, promotes the expression of *Myc*, *DNMT3B*, and *EZH2*, inducing both H3K27me3 and DNA methylation on the let-7 promoter [44,45]. Elevated levels of H3K9me3 have been linked to tumor stage, the recurrence of gastric cancer (GC), and a worse prognosis in a cohort of 261 GC patients [46].

Recently, Reyes et al. (2021), conducted a meta-analysis study describing 10 HMTs (*PRDM14*, *PRDM9*, *SUV39H2*, *NSD2*, *SMYD5*, *SETDB1*, *PRDM12*, *SUV39H1*, *NSD3*, and *EHMT2*) harboring various alterations in gastric adenocarcinomas, with an impact on specific residues of histone proteins. Computational analysis revealed reduced HMT *SUV39H2* expression in patients with GC progression, suggesting that HMTs could serve as potential biomarkers for treatment approaches [47]. In another study, HMT DOT1L was found to be involved in the methylation of H3K79 in patients with familial GC, suggesting a potential role in gastric carcinogenesis [48].

Histone demethylases (HDMs) exhibit the potential for both up and downregulation. Previous studies have shown that the upregulation of LSD1 suppresses p21 transcription by constraining its H3K4 promoter methylation, thereby resulting in GC progression. On the other hand, the downregulation of HDMs DPY300 and KDM5A was detected to enhance H3K4 methylation, consequently inhibiting GC’s growth [49,50]. Nishikawaji et al. (2016) demonstrated that GC cell growth and invasion undergo a marked reduction, via the knockdown of HDM SETDB2, which stimulates tumor-suppressor *CADM1* and *WWOX*-expression levels [51].

Histone acetylation is intricately linked to DNA repair, chromatin remodeling, and gene-expression regulatory mechanisms. Histone acetylases (HAT) and deacetylases (HAD) are the primary enzymes involved in the regulation of gene transcription, by altering chromatin structure and controlling the binding of specific transcription factors. Several studies have established associations between acetylation/deacetylation processes, tumor TNM staging, GC development, and poor prognosis [52,53]. Elevated HDAC2-expression levels correlate with unfavorable outcomes in gastric cancer [54]. Similarly, another study presents a significant correlation between elevated HDACs 1-3-expression levels coexisting with lymph-node invasion and poor OS in GC patients [55].

*Helicobacter pylori* induces p21WAP1/CIP1 expression via the acetylation of H4 in the p21 promoter. Hp also promotes gastric carcinogenesis through the upregulation of JMJD2B expression, which, in turn, enhances COX-2 expression via NF-κΒ and correlates with the tumor stage, GC cell expansion, and recurrence [56,57,58]. Previous studies have suggested a strong correlation between H3K9 acetylation and GC poor prognosis, whereas reduced acetylation levels on H3K9 and H4K16 histones have been associated with poorly differentiated GC tumors [53,58].

The *KAT2B* and *EP300* genes, which encode HATs, frequently undergo mutations or silencing in GC cases. Hp infection hinders HAT-p300 binding to the p27 promoter, leading to H4 hypoacetylation and ultimately to GC development [59].

Histone phosphorylation controls various cellular processes, such as cell signaling, apoptosis, chromosomal folding, compression, segregation, and DNA damage repair via kinases and phosphatases interaction [60]. Histones H3 and H4 are predominantly phosphorylated. In gastric carcinomas, H3S10 is less expressed in surgical resection margins, suggesting a significant prognostic role of phosphorylation in defining negative resection margins [61]. The Ras ERK1/2 signaling pathway also contributes to GC progression by inhibiting the phosphorylation of histone 1.4 at Serine 27 residue through Aurora B. Takahashi et al., in their cohort study involving 122 GC patients, demonstrated that the overexpression of phosphorylated histone H3 is indicative of poor prognosis [62]. H3S10 phosphorylation is catalyzed by Aurora kinase A (AURKA) [63].

AURKA is overexpressed in premalignant lesions such as gastric inflammation and intestinal metaplasia and is also negatively correlated with survival in gastric-cancer patients, suggesting a potential prognostic role [64,65,66].

Previous studies report that H3S10 phosphorylation is also induced by Hp infection and has been implicated in promoting gastric carcinogenesis [67]. Conversely, Hp infection in gastric epithelial cell lines diminishes H3S10 and H3T3 phosphorylation via a type-IV secretion system-dependent approach [68].

Histone ubiquitination in gastric carcinomas typically occurs subsequently to histone acetylation and/or methylation, or as a result of alterations in enzymes’ stability and activity, thereby exerting a synergistic effect on the cell cycle, DNA damage, and apoptosis [69]. The ubiquitination of mainly H2A and H2B histones impacts chromosome structure and can lead to degradation through the proteasome pathway [70]. H2B ubiquitination levels vary between differentiated carcinomas and the significantly lower levels detected in malignant tissues, strengthening the potential therapeutic role of the ubiquitination process in gastric cancer [71]. The ubiquitination of H2AK119 is consistently accompanied by H3K27me3 via the polycomb repressive complex 2 (PRC2) [72].

The Cullin4B-RING E3 ligase complex (CRL4B) is also capable of catalyzing H2AK119 ubiquitination and in cooperation with the PRC2 complex induces tumorigenesis [73]. The methylation of H3K4 and H3K79 histones via COMPASS and DOT1L enzymes, respectively, presupposes prior H2B ubiquitination [74]. The interplay between distinct histone modifications or the crosstalk between enzymes that modify histones requires further elucidation [75]. The observation that the expression of ubiquitinated H2B is notably reduced in gastric cancer cases suggests a potential therapeutic role for the histone-ubiquitination process [71].

Recently, a spectrum of novel histone modifications has been identified, like crotonylation, sumoylation, butyrylation, lactylation, biotinylation, neddylation, and succinylation [76,77]. In 2021, Fang et al. unveiled that during human embryonic stem cells’ meso-/endodermal differentiation, histone-crotonylation levels surged, resulting in a meso-/endoderm commitment. This finding suggests a connection between histone crotonylation and development processes [78]. Histone butyrylation and crotonylation seem to affect cancer proliferation and metastasis. Recently, the identification and analysis of the histone isobutyrylation process revealed the involvement of HAT1 and p300 [79,80,81].

SUMO family proteins catalyze the sumoylation of histone H4 and mediate transcriptional repression. H4K12 sumoylation dampens transcriptional activity by suppressing acetylation and methylation processes [82]. Contemporary investigations support that H4K12 sumoylation mediates to histone acetylation and methylation processes via p-300 and SET1/COMPASS suppression, respectively [82]. While the precise role of histone lactylation in cancer remains unknown, it is acknowledged to affect gene expression and metabolic regulation [80,81]. Less frequently observed histone modifications have been reported to interfere with the methylation or acetylation of histone proteins. Yet, the impact of these rare histone modifications in cancer biology remains unclear; although, evidence suggests their involvement in gene regulation, DNA damage repair, and metabolic regulation [76].

Metabolism reprogramming stands as a fundamental hallmark of cancer and intricately interacts with post-translational modifications and epigenetic processes via the crosstalk of signaling pathways [83,84,85]. Ulanovskaya et al. (2013), revealed that Nicotinamide N-methyltransferase (NNMT), a metabolic enzyme overexpressed in numerous human cancers, influences the methylation landscape of cancer cells [86].

The NNMT enzyme functions as a methyltransferase, catalyzing the conversion of nicotinamide to 1-methyl nicotinamide, utilizing the cofactor S-adenosyl-l-methionine (SAM) as the methyl-group donor. The removal of a methyl group converts SAM to S-adenosyl-L-homocysteine (SAH), known to inhibit methyltransferases. The NNMT enzyme appears to have the capacity to control the methylation status of cancer cells. Elevated NNMT levels have been described to correlate with the induction of EMT. They seem to alter gene expression not only via histone-dependent mechanisms but also through interactions with other proteins. Liang et al. (2017) conducted in vitro experiments on GC cell lines, demonstrating that heightened NNMT-expression levels activate TGF-β1/Smad signaling, thereby promoting the occurrence of EMT. Additionally, Wang et al. (2022) illustrated that NNMT activity mediates various intracellular processes by regulating the equilibrium of NAD+/NADH [87,88].

In gastric adenocarcinomas, NNMT is overexpressed, hinting at a potential role in GC tumorigenesis. In GC tumor cells, the elevated enzyme levels correlate positively with the TNM stage, tumor size, lymph-node infiltration, and distant metastasis, whereas reduced levels are linked to enhanced survival rates [89]. In GC cases exhibiting high expression levels of the enzyme, variations in the immune-cell composition within the tumor microenvironment suggest that NNMT could promote immune infiltration, potentially serving as a prognostic biomarker [90,91,92]. Furthermore, Wang et al. (2022), performing single-cell RNA sequencing analysis on over 95,000 cells originating from early gastric cardia adenocarcinoma cases, revealed that NNMT expression levels rose progressively throughout the malignant progression correlating with a worsened outcome [93].

### 3.3. Non-Coding RNAs

microRNAs (miRNAs), long non-coding RNAs (lncRNAs), and circular RNAs (circRNAs) play pivotal roles in various processes implicated in gastric carcinogenesis (cell cycle, apoptosis, proliferation, migration, and invasion), and correlate with chemo- or radiosensitivity [94]. miRNAs function by binding to the 3′ (UTR) of the mRNA target, resulting in gene silencing [95]. Yu et al., conducting miRNA microarray experiments on an early gastric-cancer mouse model, unveiled the significant involvement of the miR200 family in gastric carcinogenesis initiation. These miRNA molecules prove to be efficient predictors of overall survival in early gastric carcinomas (Table 2) [94,95].

lncRNAs mediate the regulation of chromatin remodeling, transcription, and post-transcriptional gene-expression processes [96]. HOTAIR, a frequently overexpressed lncRNA, may contribute to the dissemination of gastric-cancer cells. Previous studies suggest its involvement in specific signaling pathways, including Wnt/β-catenin and PI3K/Akt pathways, as well as its role in either silencing or upregulating HOXD and miR34a or miR-330 and miR-331-3p, respectively (Table 3) [105,106,107]. lncRNAs like H19, MNX1-AS1, MALAT1, HULC, and UCA1, have been characterized as playing an oncogenic role in gastric carcinomas [108], whereas others like CRNDE appear to inhibit GC development [109].

Another interesting lncRNA is the X-inactive specific transcript (XIST) that has been described to be overexpressed in various cancers, including gastric cancer. As previously described, lncRNA XIST acts via miR-101 and regulates EZH2, supporting the lncRNA-miRNA crosstalk that has been already mentioned in various cancers. In gastric cancer, XIST lncRNA-expression levels were found to be overexpressed in both GC patients and cell lines, and its upregulation was linked to tumor size, positive lymph nodes, and the TNM stage. It was also proposed that lncRNA XIST targets oncogene MACC1, which mediates the HGF/c-Met pathway, via its interaction with miR-497 and promotes gastric tumor growth and invasion [114]. Recent studies on the role of lncRNA XIST in gastric carcinogenesis have proposed an interplay between XIST lncRNA and Janus kinase 2 (JAK2) transcription factor that leads to GC cell migration [115]. Furthermore, another research study demonstrated that XIST lncRNA alters miR-132 and paxillin (PXN) expression levels. The last one is an adhesion molecule that has been linked to GC invasiveness. miR-132 targets the PXN gene, thus the interplay between XIST lncRNA and PXN leads to the regulation of the latter. Given the information provided, one might propose that XIST lncRNA could serve as a potential therapeutic target in gastric carcinomas [116].

circRNAs, as previously described, contribute to tumor development, metastasis, recurrence, and therapeutic resistance [117]. Sequencing data analysis of GC cases revealed numerous circRNA molecules with pro- or antitumor activity. Elevated levels of ciRS-7 are related to GC progression [118]. Acting as an oncogene, ciRS-7 counteracts the mirR-7-mediated inhibition of the PTEN/PI3K/AKT pathway in GC (Table 4). Recently, Lin et al. presented the correlation between circRIMS and gastric-cancer metastasis. In this study, circRIMS expression levels were significantly elevated in T3N3M0 and T3N1M0 gastric cancer cases, suggesting a potential diagnostic and therapeutic role of circRIMS in gastric carcinomas [119].

Overexpressed hsa_circ_0005092 and hsa_circ_0002647 molecules are positively correlated with recurrence-free survival (RFS) and OS in patients with gastric adenocarcinomas [120]. Given their high stability in body fluid, circular RNAs could serve as effective biomarkers for cancer diagnosis, prognosis, and monitoring responses to treatment [138]. Based on research conducted by Han et al., a notable decrease in circRNAs hsa_circ_0021087 and hsa_circ_0005051 was observed in tissue and liquid biopsies from GC patients, indicating their potential as a non-invasive diagnostic biomarker [121]. A recent meta-analysis in gastric adenocarcinomas revealed that hsa_circ_0002019 and hsa_circ_0074736 molecules could regulate post-transcriptionally the expression levels of several genes (*ERBB4, PRTG, SLITRK2,* and *GUCY1A2)* linked to gastric-cancer survival. Therefore, circRNAs could serve as prognostic biomarkers [122].

## 4. Gastric-Cancer Clinical Management

Gastric carcinomas exhibit notable inter- and intratumoral heterogeneity, which may partially be responsible for the unfavorable outcome of this condition. Understanding tumor biology is expected to enhance treatment efficacy and accelerate the discovery of novel prognostic and predictive biomarkers [139]. Radical surgery remains the main treatment approach for localized gastric cancer. Various therapeutic strategies, including perioperative chemotherapy and adjuvant chemo- or adjuvant chemoradiotherapy, have been proposed to decrease the recurrence risk and prolong lifespan [140,141,142,143]. Targeted therapy with antihuman epidermal receptor 2 (anti-HER2) and anti-vascular endothelial growth factor (anti-VEGF) perioperatively remains under investigation.

Prior studies have shown that perioperative chemotherapy, comparative to surgery alone, can improve patients’ prognosis, establishing it as a widely adopted practice in various countries. Numerous clinical trials have shown improved survival rates in stage II/III GC patients undergoing primary surgery, whilst the data about adjuvant radiotherapy remains controversial [140]. The precise role of targeted therapies (anti-HER2, anti-VEGF), along with immunotherapy using Immune Checkpoint Inhibitors (ICIs), remains unclear, with several phase-II/III clinical trials seeking clarity.

Molecular biomarkers such as HER2, MSI, and programmed cell death ligand 1 (PD-L1) could distinguish patients with metastatic disease who may benefit from targeted therapy or immunotherapy. Nonetheless, early detection methods and preventing-recurrence measures remain an urgent need. In the context of liquid biopsy, circulating molecules have the potential to predict recurrence [141,142]. Resistance to ICI treatment is another issue that remains to be addressed. A comprehensive investigation into epigenetics, metabolism, microbiomes, and the immune system is necessary to elucidate the underlying mechanisms of immune modulation and resistance [143,144,145].

## 5. Discussion

Gastric cancer is commonly diagnosed worldwide and exerts a significant impact on healthcare. The absence of specific clinical symptoms often leads to the diagnosis of the disease at late stages. Inter- and intratumoral heterogeneity contributes to poor prognosis. Given the unclear effectiveness of targeted therapies and immunotherapy, it is essential to establish reliable biomarkers for prognostic and diagnostic purposes. Epigenetics emerges as a promising field for acquiring comprehensive insights into the molecular processes underlying gastric cancer, as well as for developing potential prognostic and predictive biomarkers.

Promoter hyper- or hypomethylation is a common mechanism of tumor-suppressor gene silencing or oncogene activation, respectively. Tumor-suppressor genes, silenced by improper DNA methylation, serve as drivers in tumorigenesis. Widespread methylation in cancer cells also leads to non-core region methylation in surrounding tissues [14]. Previous studies have proposed an in vivo model for aberrant DNA methylation. According to this model, several inflammatory mediators, such as IL-1, IL-6, and TNF, activate DNMT1 and EZH2, causing dense methylation in promoter CpG regions. This process predisposes cancer development as it generates an “epigenetic field” effect even in noncancerous or precancerous cases [146,147]. However, in gastric carcinomas, hypomethylation, focused on the decreased methylation of repetitive DNA sequences (ALU, LINE1), as we progress from chronic to cancerous lesions, has also been described. This fact establishes hypomethylation as an early event in GC carcinogenesis and suggests that LINE1 hypomethylation could serve as a marker for “epigenetic field cancerization” [148].

Hp and EBV infections exert an impact on aberrant DNA methylation. As previously mentioned, EBV infection, although a rare event, triggers DNA methylation through the direct dysregulation of DNMT enzymes. On the other hand, Hp induces DNA methylation in two ways: directly, through the effect of its pathogenic product CagA protein, and indirectly, via chronic inflammation and the CpG-island methylation of tumor-suppressor genes such as CDH1, p16, and IL1β [147]. Additionally, previous studies have suggested that EBV could also trigger Hp infection, resulting in increased bacterial pathogenic CagA-protein activity [146]. Furthermore, as previously pointed out, the synergistic effects of Hp and EBV coinfection via oncoprotein gankyrin overexpression dysregulate migration, apoptotic, and DNA damage-repair pathways, boosting GC aggressiveness [149].

We are aware that *CDKN2A* and *MLH1* silencing are common epigenetic events in gastric carcinomas and are associated with cell-cycle regulation and DNA repair mechanisms (see Table 1). The methylation-induced silencing of p16 has been reported in 25–42% of gastric carcinomas, thus potentially serving as a biomarker for primary cancer diagnosis by predicting the malignancy of dysplastic lesions [150,151]. Approximately 30% of gastric carcinomas exhibit CDKN2A methylation, potentially contributing to the malignant transformation of gastric lesions [152,153]. The methylation of the *MLH1* promoter has been observed in 31–67% of gastric carcinomas and has been identified as an early event in GC cases, primarily in the papillary subtype [34]. Furthermore, in half of GC cases, promoter hypermethylation, as well as hemizygous deletion, lead to *RUNX3* tumor-suppressor gene silencing [154].

EBV infection-mediated methylation may occur in CpG islands as well as in low-CpG regions [155]. TCGA analysis has shown that EBV-positive tumors present a distinct CpG-island methylator phenotype (CIMP), harboring *CDKN2A* (p16) gene-promoter methylation but not *MLH1* methylation [6]. The CIMP-high phenotype is related to a more favorable prognosis, diffuse histological subtype, and earlier disease stages than a CIMP-negative phenotype [156]. EBV-positive gastric adenocarcinomas exhibit high immunogenicity and therefore could be utilized for additional studies on immunotherapy, leading to new potential targets for personalized treatment [157]. In EBV-positive GC cases, almost 270 genes are methylated, including *CXXC4*, *TIMP2*, and *PLXND1*, while *COL9A2*, *EYA1*, and *ZNF365* are highly methylated in both EBV-positive and EBV-negative/MSI-high subtypes. The *MLH1* gene promoter is frequently methylated (46%) in EBV-negative/MSI-high subtypes but not in EBV-positive gastric carcinomas. Moreover, DNA hypermethylation in the MSI subgroup strengthens the idea that specific “epimutations” serve as a compass for subsequent alterations [154].

In patients with primary gastric carcinomas, Laminin a4 subunit (LAMA4) expression is associated with high-grade tumors and predicts poor OS. *LAMA4* gene expression is upregulated in gastric-cancer cells through the binding of the ZEB1 (Zinc finger E-box-binding homeobox 1) transcription factor to the *LAMA4* promoter. It has been reported that *LAMA4* upregulation correlates with invasion and metastasis [22]. Additional molecules acting as transcription factors like caudal-type homeobox 1/2(CDX1/2), and Kruppel-like factor 5 (KLF5) are involved in the Sonic Hedgehog pathway (SHH). *CDX1/CDX2* genes seem to have a key role in intestinal dysplasia reprogramming. Previous studies have mentioned a remarkably high CDX2 expression but decreased CDX1/KLF5 expression in gastric tissue samples mainly due to methylation. Small molecules show antiproliferative activity in colon cancer through KLF5-expression inhibition, thus SHH implication in gastric carcinogenesis could be further evaluated for targeted therapy [158].

Kim et al. (2018) demonstrated that the hypermethylation of Insulin-like growth factor-binding protein 7 (*IGFBP7)* exon 1 induces gene downregulation, thereby supporting tumor-suppressor activities in gastric cancer. IGFBP7 protein-expression levels correlate with poor clinical outcomes, and therefore could be a potential therapeutic target, particularly in patients with poor prognosis [26].

*FBXO32* is a tumor-suppressor gene that encodes an F-box protein, constituting one of the four subunits of the ubiquitin protein ligase complex. The hypermethylation of *FBXO32* leads to the downregulation or loss of function. The reactivation of the *FBXO32* gene is suggested to possess prognostic significance and may offer therapeutic advantages [23]. The *CDH11* tumor-suppressor gene is also silenced through promoter hypermethylation in gastric adenocarcinomas and is considered a potential prognostic biomarker associated with malignant behavior [24,25].

The prognosis of advanced gastric adenocarcinomas can be predicted by evaluating the Claudin-3-promoter methylation status, which results in decreased protein levels. Conversely, claudin-3-promoter hypomethylation appears to contribute to the genesis of intestinal-type GC cases. Zhang et al. (2018) demonstrated that promoter hypermethylation and low claudin-3-expression levels indicate a poorly differentiated phenotype and a higher metastatic status of gastric carcinoma with lymph-node spread. The methylation profile of claudin-3 could serve as a prognostic/predictive biomarker as well as a promising therapeutic target for advanced GC cases [27].

Recently, Heo et al. (2023) showed that in both early- and advanced-stage primary gastric carcinomas, the Tensin 4 gene (TNS4) was overexpressed. Based on TCGA molecular subtypes, TNS4-expression levels were higher in the EBV-positive subgroup. The TNS4 gene encodes a protein located in focal adhesion sites, facilitating interaction between the cytoskeletal network and the extracellular matrix. Elevated TNS4 expression has been shown to promote the EMT process through AKT/GSK-3β signaling and is proposed to correlate with the development and progression of gastric adenomas, a process purportedly triggered by Hp infection [159].

In gastric carcinomas, the elevated methylation of the hTERT promoter has also been observed. To identify patients at risk of gastric-carcinoma development, hTERT hypermethylation is crucial. Gastric-cancer proliferation as well as distant and lymphatic metastasis are correlated to hTERT-promoter methylation, thus the latter could be used as a diagnostic marker of gastric carcinomas and classify patients who are at high risk of GC development [28,29].

The *SRBC* gene (serum-deprivation response factor-related gene product that binds to the c-kinase) encodes a protein that is downregulated in various cancer cell lines, suggesting a possible tumor-suppressor role. Normally, the protein regulates the traffic and/or budding of caveolae and plays a role in caveolae formation in a tissue-specific manner. The inactivation of *SRBC* is implicated in tumor resistance against chemotherapeutic agents such as oxaliplatin, as previously reported [160]. Research studies have reported that inactivation is achieved mainly through CpG-island DNA hypermethylation, while gene somatic mutations or biallelic deletion seem to be less frequent events. However, the DNA methylation of the *SRBC* gene promoter is not the only inactivation mechanism. Histone modifications were also found to play a role in gene inactivation. EZH2 trimethylates histone H3 at lysine 27, leading to gene-expression control and contributing to SRBC gene downregulation in gastric-cancer tissues [161].

Furthermore, the CD274 and PDCD1LG2 genes encode the immunosuppressive molecules PD-L1 and PD-L2. According to the TCGA group classification, only the EBV (+) GC molecular subtype exhibits elevated PD-L1- and/or PD-L2-expression levels. Recently, Zhu et al. (2020) have shown that the PD-L1 promoter was mostly hypermethylated in gastric carcinoma patients who have already received anti-PD-1 therapy. PD-L1 promoter methylation was elevated after therapy with pembrolizumab. However, surgery alone after recurrence did not have an impact on the methylation status of PD-L1. PD-L1 seems to exert an oncogenic activity and promotes cancer development and infiltration through RAS/MAP and AKT signaling pathways. Resistance to anti-PD-1 immunotherapy might develop because of PD-L1 promoter methylation [30].

PD-L2 is a second ligand of PD-1 and can be expressed via tumor cells. In gastric carcinomas, EBV status, CIMP phenotype, as well as *MLH1* and *CDKN2A* methylation status, were strongly associated with methylated PD-L2. The TCGA group found that PD-L2 methylation is associated with EBV infection, CD8+ T cell infiltration, microsatellite instability, and high tumor mutational load. PD-L2 hypermethylation appears to be a promising biomarker for the prediction of responses in GC patients after anti-PD-1 immunotherapy [31].

DNMTs are pivotal contributors to the process of DNA methylation. Previous studies have shown elevated levels of DNMT-1- and DNMTs-3A/3B-protein expression in both HP-induced gastritis and gastric carcinomas [162,163]. Hedayati et al. (2022), presented a correlation between Hp pathogenicity and increased DNMT1-expression levels [164], whereas Song et al. (2020) identified a novel epigenetic signature in gastrointestinal adenocarcinomas. They scrutinized their findings along with TCGA datasets for DNA methylation and RNA sequencing and identified a total of five methylation-driven genes, capable of predicting overall survival (OS) in GC patients. Three of those, *HENMT1*, *GRIN2A*, and *STC2,* seem to relate to processes such as hypoxia response and hormone-metabolism regulation, thereby contributing to the development and progression of several carcinomas, including gastric adenocarcinomas [165].

Purkait et al. (2020), have documented the interplay between DNA methylation and histone modifications in the pathogenesis of gastric carcinomas. DNMT-1/3A/3B and EZH2 expression were significantly upregulated in Hp-associated gastritis and carcinomas. EZH2-expression levels were notably higher in cases of metaplasia and exhibited a positive correlation with DNMT-expression levels. Epigenetic modifications can be reversed, thus targeting DNMTs and EZH2 could offer a promising therapeutic approach for gastric carcinoma [163]. Numerous inhibitors targeting DNMTs and EZH2 are examined to assess their potential efficacy in treating various malignancies [166,167].

Takeshima et al. revealed that chromatin remodelers like SMARCA1 present a distinct methylation status in GC patients, compared to noncancerous tissues. That claim it itself supports that chromatin remodeler disorders occur early during tumorigenesis and contribute to the production of an epigenetic field effect [168].

Numerous in vitro and in vivo studies have suggested that Hp infection causes DNA double-strand breaks (DSBs) and activates the ATM (Ataxia-Telangiectasia-Mutated serine/threonine kinase) response. ATM is involved in the DSB-repair pathway and phosphorylates other DSB-implicated proteins. Santos et al. (2018) demonstrated that *ATM* gene transcription is epigenetically regulated via promoter hypomethylation as well as the hyperacetylation of H3/H4 histones. Chromatin immunoprecipitation-quantitative PCR experiments unveiled that H3 and H4 histones were highly acetylated four hours after Hp infection [169].

In certain cancer types, the chromatin landscape of various genes and their interactions with polycomb repressive complex 2 (PRC2) and acetyltransferase complexes can significantly influence the transcriptional status of the entire genome. It is well recognized that histone modifications including H3/H4 acetylations and H3 trimethylation are linked to gene upregulation, while gene downregulation is mainly achieved through H3 lysine 9 di-/trimethylation as well as H3 lysine 27 trimethylation [146]. Members of the MLLs family serve as tumor-suppressor genes and mediate in gene activation through histone H3 methylation at lysine 4 amino acid (H3K4). Lysine-specific demethylase 6A (KDM6A) also regulates gene expression through H3K27 demethylation. Previous studies in GC cases have highlighted the association between PCAF (P300/CBP-associated factor) loss and poor outcomes. Brasacchio et al. (2018), have demonstrated that PCAF loss has an impact on gastric adenocarcinoma initiation, thus PCAF could serve as a candidate acetylation factor [32,33].

Hp and EBV infections alter histone modifications as previously described. EBV infection converts H3K9me3+ heterochromatin into a state characterized by H3K4me1+/H3K27ac+ bivalency. This fact leads to the activation of latent enhancers that stimulate the expression of genes implicated in gastric carcinogenesis like *TGFBR2* and *MZT1* [170]. Hp upregulates the expression of both the *p21 WAP/CIP1* tumor-suppressor gene as well as the *JMJD2B* gene, which promotes GC carcinogenesis through histone acetylation.

Non-coding RNAs, miRNAs, and circRNAs have been extensively studied as one of the three pillars of epigenetics. Fan et al. (2020) elucidated elevated expression levels of the *CCDC144NL-AS1* gene, which are related to unfavorable prognosis, invasion, migration, and apoptosis inhibition in GC cells [110]. CCDC144NL-AS1 likely acts as a competing endogenous RNA (ceRNA), occupying common miRNA binding regions. Therefore, it could serve as a promising diagnostic and therapeutic target [171]. HOTAIR has been described as being overexpressed in gastric cancer, playing a pivotal role in metastasis and survival. The specific lncRNA molecule engages in the interplay with miRNAs and acts as a sponge for miR331-3p. Moreover, it can trigger the HER2 target’s expression levels [172].

Liu et al. (2021), analyzed LINC01232 in gastric-cancer cell lines and the TCGA dataset. Their findings unveiled that KLF2 expression is inhibited through H3K27me3 histone methylation. The knockdown of LINC01232 is shown to suppress the proliferation of gastric-cancer cells both in vivo and in vitro. This finding suggests LINC01232’s potential as an effective therapeutic target for gastric cancer (Table 2) [111].

Recently, Peng et al. (2022), performing RNA-Seq data analysis in stomach adenocarcinomas, demonstrated that the lncRNA TM4SF1-AS1 inhibits the immune killing capacity mediated by T cells and serves as a prognostic marker for assessing the immune response to anti-PD1 therapy [112].

The zinc finger antisense 1 (ZFAS1) is another lncRNA demonstrating oncogenic activity in gastric cancer. RNA sequencing analysis has revealed that upregulated ZFAS1 expression correlates with diminished levels of hypoxia-inducible factors 1 and 2 (HIF1, HIF2) suggesting a potential involvement of ZFAS1 in HIF1A epigenetic silencing. ZFAS1 upregulates the HIF2 protein levels under both hypoxia and normoxia conditions. The knockdown of ZFAS1 via siRNA in gastric-cancer cell lines results in decreased cell migration, invasion, and proliferation, providing evidence that ZFAS1 may serve as an independent prognostic biomarker as well as a therapeutic target in gastric cancer [113].

Lately, a comprehensive analysis of immune-related long non-coding RNAs (Jin et al., 2023) identified nine significant lncRNA molecules. Gastric cancer was stratified into five subtypes, with cluster C3 being the most versus cluster C5 being the least immunogenic. This study documented that the specific lncRNA subtypes possess the ability to identify patients who may derive benefit from receiving chemotherapy and immunotherapy [173].

Regarding miRNAs in gastric-cancer development, their role has been extensively examined. Data analysis conducted by Kipkeeva et al. (2020), revealed that miRNAs associated with the spread of cancer cells are commonly engaged with the Wnt/-catenin pathway or influence genes that trigger the EMT process through interaction with other pathways. miRNAs implicated in chemotherapy resistance mostly target apoptotic regulators and are linked to the PI3K/AKT/mTOR pathway. Their potential as biomarkers holds considerable promise for detecting and monitoring gastric cancer [174].

As previously described, the direct targeting of the nuclear protein Forkhead box M1 (FoxM1) triggers pathways such as MEK/ERK, NF-κB, and PI3K/AKT, via mir-320d in GC cases, suggesting its tumor-suppressive properties and positioning mir-320d as a potential biomarker for cancer prognosis and treatment (Table 3) [97]. Additionally, miR-942 is implicated in invasion and migration processes. miR 942 3p functions as an oncogene, while miR 942 5p acts as a tumor-suppressor gene. Moreover, serum miR-942 expression levels are linked to disease progression and low survival rates, suggesting its utility as a prognostic biomarker [98].

miR-141 and miR-1269b are also implicated in gastric carcinomas. The first one affects MEK/ERK- and MAPK-signaling pathways and has been demonstrated to inhibit proliferation and trigger apoptosis in gastric-adenocarcinoma cell lines. miR-1269b, through its regulatory control over methyltransferase-like 3 (METTL3), exerts inhibitory effects on gastric-cancer development. The overexpression of miR-1269b suppresses the migration and invasion of GC cells, whereas its inhibition leads to the opposite effects [99,175]. Reduced levels of miR-203a/b, accompanied by the methylation of their proximal promoters, indicate their role as tumor-suppressive microRNAs in GC cases. Furthermore, miR-3196, miR-1244, miR-135b-5p, and miR-628-3p are associated with GC differentiation, while miR-196a-5p appears to correlate with the age of GC onset.

In another study, the performance of a miRNA microarray analysis on 353 Japanese patients with primary gastric tumors identified a GC miRNA signature consisting of 22 up- and 13 downregulated miRNAs. In this study, a “histotype” miRNA panel was developed, capable of distinguishing diffuse and intestinal-type gastric tumors. Moreover, miR-let7g emerged as an independent predictive biomarker for disease-free survival (DFS) [176]. The let-7 family negatively regulates *HMGA2*, whose elevated expression levels are associated with tumor invasiveness and unfavorable outcomes, suggesting a potential prognostic role in gastric cancer [104].

Qu et al. (2020), utilizing miRNA-expression data from 386 GC cases reported from the TCGA group, demonstrated that miR-30a-3p and miR-105-5p have the potential to serve as biomarkers for MSI-H gastric adenocarcinoma, likely due to their ability to modify the expression of DNA damage-repair genes [102]. The GEO database analysis of miRNA- and mRNA-expression data from GC cases identified hsa-miR-196b-3p and four important nodal genes (*CALML4*, *SMAD6*, *PITX2*, and *TGFB2*) as prognostic GC biomarkers in Hp-positive cases. It was suggested that hsa-miR-196b-3p may serve as a reliable biomarker for predicting GC prognosis [100].

Gilani et al. (2022) analyzed the GSE106817 dataset with 2.566 miRNAs, utilizing the Boruta machine learning variable-selection approach, and discovered a total of 30 miRNAs capable of serving as biomarkers in diagnosing GC. hsa-miR-1343-3p presented the highest ranking among them. Utilizing artificial intelligence technology, they identified hsa-miR-1343-3 as a strong candidate for biomarker analysis in GC diagnosis, as well as hsa-miR-8073 and hsa-miR-1228-5p, which exhibit significant contributions to GC prediction [101].

Additionally, miR-1228 via the downregulation of the macrophage migration inhibitory factor (MIF) serves as a negative regulator of gastric-cancer growth and angiogenesis, supporting its use as a potential therapeutic target for anti-angiogenic therapy against gastric cancer [103].

Concerning the role of circular RNAs, numerous studies have been conducted. Currently, traditional tumor markers like CEA and CA19-9 have limited clinical utility due to their decreased sensitivity and specificity. Research studies have established the presence of circRNAs not only within tissues but also in human serum, plasma, and other body fluids [177].

Previous studies have highlighted elevated levels of CircAKT3 and circLMO7 in gastric-cancer cells, suggesting potential implications in GC development [123]. The elevated expression levels of serum circSHKBP1 (hsa_circ_0000936) are associated with diminished survival rates and advanced TNM stages, as reported by Xie et al. (2020) (Table 4). Furthermore, the plasma levels of hsa_circ_0000745 correlate with TNM staging. The study conducted by Reisdas-Mercês et al. (2022) demonstrated that hsa_circ_0000211, hsa_circ_0000284, and hsa_circ_0004771 maintain consistent expression profiles across various techniques (RNA-Seq and RTqPCR) and distinct sample types (tissue and blood) [124,178].

Several circRNAs exhibit potential for early gastric-cancer screening in combination with other tumor markers, though their sensitivity remains limited when used alone. For instance, the co-detection of hsa_circ_0001017 and hsa_circ_0061276 in both gastric-cancer tissues and patient plasma exhibits a significant diagnostic potential, with a sensitivity and specificity of 95.5% and 95.7%, respectively [179].

Zheng et al. (2022) observed a significant reduction in the expression levels of exosomal hsa_circ_0015286 in GC patients shortly after the surgical treatment. This noteworthy finding indicates that exosomal hsa_circ_0015286 has the potential to serve as a non-invasive biomarker for the diagnosis and prognostic evaluation of GC [125].

Zhang et al. (2017) demonstrated that the decreased expression of circLARP4 in GC tissues serves as an independent prognostic indicator for the overall survival of GC patients [126].

The differential expression of serum hsa_circ_0007507 among post-operative GC, gastritis, intestinal metaplasia, and relapsed patients indicates its potential utility as a novel diagnostic and dynamic-monitoring biomarker for GC [127]. Significantly distinct levels of hsa_-circ_002059 were also observed in plasma samples collected post-operatively compared to those collected pre-operatively. The lower expression levels exhibited a significant association with distant metastasis, the TNM stage, and patient age [128]. Prior studies have demonstrated a close relation between hsa_circ_0000467-expression levels and TNM staging, while the downregulation of hsa_circ_KIAA1244 in GC patients suggests a possible role as a diagnostic marker for GC, considering its correlation to TNM stage, metastatic potential, and shorter survival rates [129].

Zhang et al. identified several circRNAs (circDLST, circCACTIN, and circNRIP1) pivotal in tumor growth, migration, and invasion [130,131,132]. Wang et al. (2019), reported elevated levels of CircLMTK2 in GC tissues, with its presence correlating with unfavorable prognosis and advanced TNM stages. circOSBPL10 (hsa_circ_0008549) exhibits a significant upregulation in GC tissues and promotes gastric-cancer cell growth. Therefore, circOSBPL10 (hsa_circ_0008549) was proposed as a novel prognostic biomarker in gastric carcinomas [133,134].

CircRNAs can also impact drug resistance. Shi and Wang (2022) demonstrated that Circ_AKT3 knockdown results in increased cisplatin sensitivity in cisplatin-resistant GC cells via the miR-206/PTPN14 axis. Furthermore, Fan et al. (2022) found that the METTL14-mediated m6A alteration of circORC5 inhibits gastric-cancer progression. Fang et al. (2022) discovered that circCPM plays a crucial role in regulating GC autophagy and 5-Fluoro- Uracil resistance, while Chen et al. (2021) revealed that CircDLG1 was highly increased in distant metastatic lesions and anti-PD-1-resistant gastric-cancer tissues. CircDLG1 was also linked with an aggressive tumor phenotype and poor prognosis in GC patients treated with anti-PD-1 drugs [135,136,137,180].

The linkage between DNA methylation and miRNAs has been characterized in an expanded mass of the literature as an additional tier in the control of gene expression. Several studies have mentioned the crosstalk between DNA methylation and miRNAs. This interplay happens via the methylation process of miRNA promoters leading to the control of their expression, or through the inhibition of enzymes (Dicer, Drosha) that participate in the miRNA process. On the other hand, miRNA molecules alter gene methylation profiles by modifying enzymes or co-factors that take part in the DNA methylation process [181]. A study conducted in GC patients (2016) reported that miR-106a promoter hypomethylation resulted in its overexpression, and plasma miR-106a expression levels were decreased after gastrectomy, illustrating a potential diagnostic role for miR-106a [182].

Another interesting interaction is the one between miRNAs and histone-modifying enzymes. Previous studies report direct and indirect interactions with enzymes or mechanisms mediating chromatin remodeling. The major components of the miRNA-chromatin-remodeling network include signaling pathways, transcription factors, DNA methylation, long non-coding RNAs, and various chromatin-remodeling complexes. These elements are crucial for the regulation of processes such as apoptosis, cell proliferation, and differentiation [183].

Epigenetic drugs are compounds that target and repair various post-translational modifications of enzymes implicated in histone remodeling and DNA methylation processes [184]. They are classified into five groups: DNA methyltransferase inhibitors (iDNMTs), histone methyltransferase inhibitors (iHMTs), histone demethylase inhibitors (iHDMs), histone acetyltransferase/de-acetyltransferase inhibitors, (iHATs/iHDACs), and miRNA inhibitors (anti-miRs). However, only few iDNMTs and iHDACs have been approved by the FDA for use in several malignancies including cancer. Both iHMTs and iHDMs as well as anti-miRs are under investigation and numerous clinical studies are still ongoing [185].

Epigenetics and epigenetic treatment options represent rapidly advancing fields. Novel pharmaceutical compounds that address pharmacokinetic and instability problems in epigenetic treatment hold promise for surpassing previous generations of DNMT and HDAC inhibitors [186]. In addition, recent research has assessed the synergistic effects of iDNMTs and iHDACs, which provide further support for the encouraging findings. However, the utilization of these compounds in clinical practice is diminishing due to their lack of specificity and potential to induce mutations that contribute to additional carcinogenesis [185]. Therefore, the pharmaceutical research focuses not only on the development of next-generation iDNMTs and iHDACs, but also in other compounds such as NNMT inhibitors and anti-miRs. 

Lately, Gao et al. (2021) investigated NNMT inhibitors and discovered the GYZ-319 compound as a potent iNNMT [187]. However, when it was used against several cancer cell lines, it showed poor cell permeability. The lack of cellular activity was rendered to the presence of two highly polar functional groups, which are found in all effective bisubstrate NNMT inhibitors: the carboxylic acid and amine groups of the amino acid motif. Therefore, van Haren et al., from the previous study group, conducted further research and utilizing the RaPID mRNA display methodology identified a group of macrocyclic peptides with NNMT-affinity properties [188]. Van Haren et al. (2021) conducted a study where they chemically modified the iNNMT GYZ-319 molecule to enhance its cellular activity. The process included converting carboxylic acid into several esters, followed by their assessment. Their experiments resulted in the conclusion that the isopropyl ester **12e** and the isopropyl ester/TML dual prodrug **14e** were the most promising molecules according to stability and cellular activity evaluation [189].

In recent years, several studies have focused on the development of epigenome-targeted therapies. Moro et al. (2020) examined epigenetic priming in multiple gastric cell lines, evaluating the efficacy of SN38, CDDP, PTX, and 5-FU cytotoxic drugs for gastric cancer. Their findings demonstrated that epigenetic priming is effective in cell lines resistant to SN38 and CDDP, as well as that genes involved in apoptosis and cell death were activated [190].

Furthermore, another study conducted by Hu et al. (2022) revealed that nanoparticles could be produced in a manner that could be able to deliver epigenetic drugs, suggesting a possible therapeutic tool in epigenome therapy [191].

Additional studies confirm that epigenetic agents, in conjunction with specific compounds, may boost antitumor immunity. Ongoing clinical studies investigate the combined use of epigenetic drugs and immune checkpoint inhibitors to evaluate their therapeutic benefits and potential adverse effects [192]

Finally, understanding the role of miRNAs in gastric-cancer epigenetics has spurred the development of anticancer miRNA-replacement therapy. miRNAs exhibiting oncogenic activity are considered therapeutic targets. Anti-miR oligonucleotides could be used to bind the guide strand of miRNAs. Thus, a multiple-miRNA-inhibition therapy in combination with other chemical compounds could achieve a synergistic antitumor effect [193].

## 6. Conclusions

Given the intricate nature of the mechanisms underlying the onset and progression of gastric carcinomas, identifying pivotal contributors to cancer pathogenesis holds immense significance. It is imperative to comprehend the precise mechanisms that underlie significant changes in epigenetic drivers. The identification and application of non-invasive epigenetic biomarkers will ensure the early detection and monitoring of both early- and advanced-stage gastric-cancer patients, facilitating tailored treatment strategies based on their epigenome profile. However, so far, we are lacking a specific and validated epigenetic signature. Therefore, numerous clinical trials involving the use of various epigenetic-modulating agents alone or in a multi-therapy approach with encouraging outcomes are ongoing. Deciphering the epigenetic network provides tremendous potential for precision medicine and will allow the implementation of revolutionary epigenetic approaches (anti-miR therapy and nanoparticles delivering epigenetic drugs) in routine clinical practice.

## Figures and Tables

**Figure 1 ijms-25-03381-f001:**
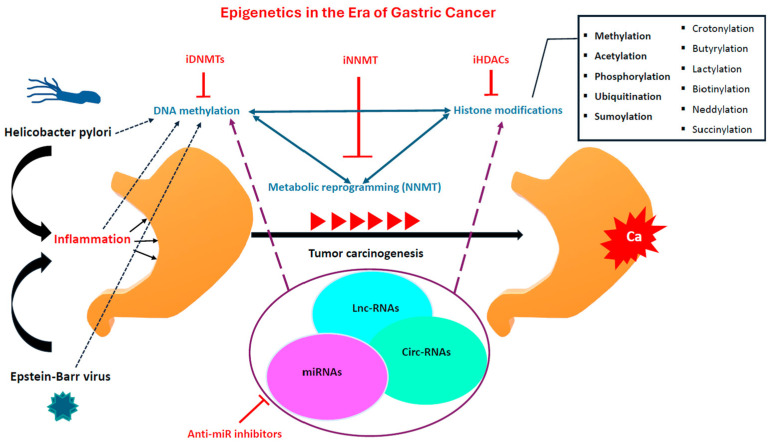
The main epigenetic mechanisms and environmental factors that mediate during gastric carcinogenesis; anti-miR inhibitors: anti-microRNA inhibitors; circ-RNAs: circular RNAs; iDNMTs: DNA Methyltransferase inhibitors; iHDACs: Histone deacetylase inhibitors; iNNMT: Nicotinamide N-methyltransferase inhibitors; lnc-RNAS: long non-coding RNAs; miRNAs: micro-RNAs; NNMT: Nicotinamide N-methyltransferase.

**Table 1 ijms-25-03381-t001:** Common epigenetically altered genes through DNA methylation in gastric carcinomas.

Gene	Role	Expression	Role in Gastric Carcinomas	References
*CDH1*	Tumor-suppressor gene	Silenced	- Elevated GC risk; - Associated with worse OS and DFS. Potential therapeutic biomarker.	[18]
*MLH1*	Mismatch-repair mechanism	Silenced	- Better prognosis for resectable GC tumors; - Oxaliplatin resistance in GC patients. Potential therapeutic biomarker.	[6,19]
*CDKN2A*	Cell-cycle arrest	Silenced	GC development through silencing mediated by Hp and EBV infections. Potential therapeutic target.	[20,21]
*LAMA4*	Encodes laminin subunit alpha 4, a member of extracellular matrix glycoproteins	Overexpressed	- Poor OS; - Increased invasion and metastasis. Potential prognostic biomarker.	[22]
*FBX032*	Tumor-suppressor gene mediates in cell-survival regulation	Downregulated or loss of function	Predicts metastasis and poor prognosis in stage-III and -IV gastric-cancer patients. Potential prognostic and therapeutic biomarker.	[23]
*CDH11*	Tumor-suppressor gene	Silenced	Potential prognostic biomarker of malignant behavior.	[24,25]
*IGFBP7*	Regulation of insulin-like growth factors (IGFs)—potential tumor suppressor gene	Silenced	Suppressive effect on gastric-cancer development when it is expressed. Potential prognostic and therapeutic biomarker.	[26]
*Claudin-3*	Cell-adhesion molecule	Downregulated	Predictor of high metastatic status and LN spread. Potential prognostic and therapeutic biomarker.	[27]
*hTERT*	Part of the telomerase complex—mediates in cellular immortalization	Overexpressed	Poor prognosis and shorter OS in GC patients. Potential diagnostic and prognostic biomarker.	[28,29]
*PD-L1*	Immunosuppressive molecule—acts as an oncogene	Silenced	Resistance to immunotherapy. Potential therapeutic biomarker.	[30]
*PD-L2*	Immunosuppressive molecule	Overexpressed	Predictor of response after PD-L1 therapy. Potential predictive biomarker.	[31]
*SRBC*	Tumor-suppressor gene	Downregulated	Chemoresistance (against oxaliplatin). Potential therapeutic target.	[30]
*PCAF*		Loss of function	Poor outcomes. Potential prognostic biomarker.	[32,33]

DFS: disease-free survival; GC: gastric cancer; LN: lymph node; OS: overall survival.

**Table 2 ijms-25-03381-t002:** miRNAs as candidate biomarkers in gastric cancer.

Micro-RNA	Action	Role in Gastric Carcinomas	References
miRNA200	Promotes oncogenesis	Predictor of OS Prognostic biomarker	[95,96]
miR-320d	Tumor suppression	Treatment of GC Prognostic and therapeutic biomarker	[97]
miR-942-3p/-5p	Oncogenic/tumor suppression	Potential prognostic biomarker	[98]
miR-141	Inhibits the proliferation of cancerous cells and triggers apoptosis	Predictor of OS Potential therapeutic biomarker	[98,99]
miR-1269b	Inhibits the development of GC, suppresses migration and invasion	Predictor of OS Potential diagnostic and prognostic biomarker	[98,99]
miR-203 a/b	Tumor suppressor	Potential prognostic biomarker	[98,99]
miR-196b-3p	Associates with the age of onset	Potential prognostic biomarker	[100,101]
miR-30-3p/miR-105-5p	Modifies expression of DNA damage-repair genes in MSI-H tumors	Predictive biomarkers for microsatellite instability	[102]
miR-1343-3p	Tumor suppressor Antiangiogenic role	Potential diagnostic biomarker	[101]
miR-8073	Tumor suppressor	Potential diagnostic biomarker	[101]
miR-1228-5p	Negative regulator of gastric-cancer growth and angiogenesis	Potential diagnostic biomarker and therapeutic target for anti-angiogenic therapy	[103]
miR-let7g	Predictive biomarker of DFS	Potential prognostic biomarker	[104]

DFS: disease-free survival; GC: gastric cancer; OS: overall survival.

**Table 3 ijms-25-03381-t003:** Lnc-RNAs as candidate biomarkers in gastric cancer.

LncRNA	Action	Role in Gastric Carcinomas	References
HOTAIR	Oncogenic	- associated with the incidence of venous invasion and poor prognosis in diffuse gastric carcinomas; - regulates cisplatin resistance. potential therapeutic target	[106,107]
H19	Oncogenic	- modulates proliferation and immune escape of GC potential therapeutic target	[108]
CRNDE	Tumor suppressor	- induces cisplatin resistance potential therapeutic target	[109]
CCDC144NL-AS1	Acts as competing endogenous RNA	- inhibition of apoptosis, invasion, migration, poor prognosis potential prognostic biomarker	[110]
LINC01232	Oncogenic	- promotes GC proliferation potential therapeutic target	[111]
TM4SF1-AS1	Involved in the tumor’s immune microenvironment	- prognostic marker for the immune response towards anti-PD1 therapy	[112]
ZFAS1	Oncogenic	- independent prognostic biomarker as well as a therapeutic target in gastric cancer	[113]
XIST	Acts as competing endogenous RNA	- promotes GC proliferation and migration potential prognostic biomarker and therapeutic target	[114]

GC: gastric cancer.

**Table 4 ijms-25-03381-t004:** circ-RNAs as candidate biomarkers in gastric cancer.

Circ-RNA	Effect	Role in Gastric Carcinomas	References
ciRs-7	GC progression	Prospective prognostic and therapeutic biomarker	[118]
circRIMS	Predicts invasive metastasis	Potential diagnostic and therapeutic biomarker	[119]
hsa_circ_0005092/hsa_circ_0002647	Upregulated	Prognostic and predictive for post-operative recurrence biomarker	[120]
hsa_circ_0021087/hsa_circ_0005051	Occurrence and development of GC	Non-invasive diagnostic biomarkers	[121]
hsa_circ_0002019/hsa_circ_00074736	Regulates the expression of genes linked to GC survival	Predictors of OS Potential prognostic biomarker	[122]
CircAKT3/circLM07	GC progression	Predictor of OS Potential prognostic biomarker	[123]
CircSHKBP1	GC progression, poor survival	Potential non-invasive diagnostic and prognostic biomarker	[124]
has_circ_0015286	Non-invasive diagnostic biomarker	Potential diagnostic and prognostic biomarker	[125]
circLARP4	Tumor suppressor	Prognostic factor for OS	[126]
hsa_circ_0007507	Differentially expressed in GC patients, post-operative GC patients, gastritis patients, intestinal metaplasia patients	Potential diagnostic and monitoring biomarker	[127]
hsa_circ_002059	Correlation with TNM stage, distant metastasis, and age of onset	Potential diagnostic biomarker	[128]
hsa_circ_0000467/hsa_circ_KIAA1244	Correlation with TNM and metastasis	Potential prognostic biomarker	[129]
circDLST, circCACTIN, circNRIP1	Promote oncogenesis, migration, and invasion	Potential prognostic biomarker	[130,131,132]
CircLMTK2	Correlation with TNM	Potential prognostic and therapeutic biomarker	[133]
CircOSBPL10	Promotes tumor growth	Potential prognostic biomarker	[134]
Circ_AKT3	Association with cisplatin sensitivity	Potential therapeutic biomarker	[135]
CircCPM	Regulates autophagy and 5-Fluro-Uracil resistance	Potential therapeutic biomarker	[136]
CircDLG1	Increases distant metastasis, anti-PD-L1 resistance	Predictor of OS Potential therapeutic biomarker	[137]

GC: gastric cancer; OS: overall survival; TNM: TNM classification of malignant tumors.

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
