# Peer review of "Gastric Cancer in the Era of Epigenetics"

_ijms, 2024, doi:10.3390/ijms25063381_

Round 1
Reviewer 1 Report
Comments and Suggestions for Authors
The manuscript “Gastric Cancer in the Era of Epigenetics” by Christodoulidis et al. is a review article about the epigenetic factors and mechanism that are involved in gastric cancer, for diagnosis, prognosis or that may be useful therapeutic targets. The manuscript might be of interest for the readers. However, there are important flaws. Authors are requested to address the following concerns since, as it is, the manuscript cannot be accepted for publication.
Major
1. It seems that the manuscript was written carelessly. There are many things to improve. For instance, the manuscript has two paragraphs with number 5; thus authors did not review the manuscript before submitting.
2. The abstract is absolutely unacceptable. It is a collection of unlinked sentences that must be completely rewritten. Please provide a proper background and identify the focus and aim of the review.
3. A figure summarizing the main epigenetic mechanisms involved in gastric cancer is necessary since it will help the readers to focus on the theme.
4. In table 1 I do not understand why authors sometimes use the word “upregulated” and sometimes “overexpressed”. Please reconcile.
5. Tables 3 and 4 should be improved. Authors use the word “biomarker” which does not explain if it is a diagnostic biomarker, or a prognostic biomarker. Instead, this is specified in table 2. Moreover there are many inconsistencies: for instance, in table 4 is written “GC progession” and sometimes just “cancer progression”. This confuses the reader.
6. The paragraph histone modifications covers only partially the available literature.
7. The manuscript lacks of important studies available in literature. A major player in gastric cancer progression is the enzyme nicotinamide N-methyltransferase (NNMT) which has been proven to be upregulated in gastric cancer (PMID: 36139012), and whose expression is correlated to a worse prognosis (PMID: 27152242; PMID: 36977555) and correlated to EMT transition in gastric cancer (PMID: 29541230). Since NNMT can affect NAD homeostasis, NAD-dependent enzymes and concentration of SAM, it has a great impact on epigenetics, as demonstrated by Ulanovskaya et al. in an elegant study (PMID: 23455543).
A number of NNMT inhibitors are already available and could be tested for gastric cancer management (PMID: 34572571; PMID: 34704059; PMID: 34424711). All these considerations cannot be ignored in a review article regarding epigenetic mechanisms in gastric cancer.
Comments on the Quality of English LanguageModerate editing of English language required
Reviewer 2 Report
Comments and Suggestions for Authors
The PubMed searches could have been with boolean expressions
What specific AND, OR and NOT could have been used to filter. Was there specific PRISMA guidelines
While mentioning lncRNAs, pl specific role of XIST in gastric cancers
A lot of text is verbatim and taken directly, so pl check plagiarism
Pl find attached the word document with some comments
Is there a truthset the authors want to describe on candidate genes?
Scores on a scale of 0-5 with 5 being the best
Language : 2
Novelty: 3
Brevity: 3
Scope and relevance: 3

Reviewer 3 Report
Comments and Suggestions for Authors
There are many good reviews in this field to name a few:
PMID: 15819717
PMID: 15930038
PMID: 17394762
PMID: 22796521
PMID: 26823082
PMID: 27718135
PMID: 28513632
Considering this situation, it may be doubtful the readers pay attention to this article particularly.
I am not sure the authors intentionally do not mention about these, but anyway it is not balanced. The section of non coding RNA and other aspects are informative as teaching materials, but the readers would wonder it is particularly important in gastric cancer or it is the case in cancers in general.
1. The discussion is only as to advanced cancer of the stomach, then the important role at initiation step of gastric cancer should be added.
PMID: 22761333
PMID: 24744581
PMID: 11350605
PMID: 19671196
2. The authors claim that epigenetic changes could be a therapeutic target (such as IGFBP; but the situation is known in many genes even such as MLH1; in terms of methylation and downregulation). The genes epigenetically changed would be candidates anyway, but the agents altering epigenetic change (authors addressed in the last section Line 539 -) has larger problems as to specificity. Expand this issue.
Round 2
Reviewer 1 Report
Comments and Suggestions for Authors
Authors addressed all the concerns and thus the manuscript can be accepted for publication.
Comments on the Quality of English LanguageMinor editing of English language required
Author Response
Thank you for your comments! We performed a minor editing of English language
Reviewer 2 Report
Comments and Suggestions for Authors
I am satisfied with the changes rendered
Comments on the Quality of English LanguageI am satisfied with the changes rendered
Author Response
Thank you for your comments!
Reviewer 3 Report
Comments and Suggestions for Authors
The revisions are acceptable.
Author Response
Thank you for your comments!